# Degradation Monitoring in Reinforced Concrete with 3D Localization of Rebar Corrosion and Related Concrete Cracking

**Charlotte Van Steen** * and **Els Verstrynge** *

Materials and Constructions Section, Department of Civil Engineering, KU Leuven, 3001 Leuven, Belgium
* Correspondence: charlotte.vansteen@kuleuven.be (C.V.S.); els.verstrynge@kuleuven.be (E.V.)

**Abstract:** Corrosion of the reinforcement is a major degradation mechanism affecting durability and safety of reinforced concrete (RC) structures. As the corrosion process starts internally, it can take years before visual damage can be noticed on the surface, resulting in an overall degraded condition and leading to large financial costs for maintenance and repair. The acoustic emission (AE) technique enables the continuous monitoring of the progress of internal cracking in a non-invasive way. However, as RC is a heterogeneous material, reliable damage detection and localization remains challenging. This paper presents extensive experimental research aiming at localizing internal damage in RC during the corrosion process. Results of corrosion damage monitoring with AE are presented and validated on three sample scales: small mortar samples (scale 1), RC prisms (scale 2), and RC beams (scale 3). For each scale, the corrosion process was accelerated by imposing a direct current. It is found that the AE technique can detect damage earlier than visual inspection. However, dedicated filtering is necessary to reliably localize AE events. Therefore, AE signals were filtered by a newly developed post-processing protocol which significantly improves the localization results. On the smallest scale, results were confirmed with 3D micro-CT imaging, whereas on scales 2 and 3, results were compared with surface crack width measurements and resulting rebar corrosion levels.

**Keywords:** acoustic emission; concrete; corrosion; damage detection; localization

## 1. Introduction

The increasing amount of deteriorating infrastructure demands for effective methods to obtain an enhanced and early assessment of the damage level to reliably decide on maintenance and repair interventions. Therefore, by combining certain non-destructive techniques (NDT) and structural health monitoring (SHM), the structural integrity of different parts of the structure can be assessed non-invasively. Some techniques would even allow the detection of damage that is not yet visible for the human eye. Specifically for damage mechanisms that start internally, such as corrosion in reinforced concrete (RC), such techniques are particularly interesting.

Corrosion of the reinforcement is generally found to be the most common deterioration problem in existing RC structures. Due to the high alkaline environment of the surrounding concrete, the reinforcement is normally protected from corrosion. However, the protective oxide layer is easily destroyed after which a complex electrochemical process takes place. At anodic sites, steel dissolves which causes a reduction of the cross section of the rebar. At cathodic sites, iron ions combine with water and oxygen forming corrosion products or rust.

The corrosion process will influence the behavior of the steel, the concrete, and the interface. First, as previously mentioned, corrosion will lead to a reduction of the cross section of the rebar. Contradicting findings are reported in the literature on the effect of corrosion on the yield stress and tensile strength [1–4]. However, authors have reached the consensus that corrosion causes a drastic reduction of the ductility of the steel [3–6]. The rebar will undergo less plastic deformation before rupture. Second, as the formed corrosion

products have a lower density, and thus occupy a greater volume than the original steel volume, they exert an internal pressure on the surrounding concrete. Eventually, the tensile strength of concrete is reached due to this pressure build-up which will result in concrete cover cracking, delamination, and even spalling of the concrete cover. Third, cracking and steel section loss will have an influence on the bond between steel and concrete. A small amount of corrosion (0–4% mass loss) is found to have a positive effect on the bond strength as corrosion products fill the pores closest to the rebar leading to more confinement [7,8]. For higher corrosion levels, the bond strength decreases. This is mainly caused by the formation of cracks, but also the reduction of the mechanical interlocking plays an important role [9]. These three mechanisms affect the eventual load-bearing capacity of RC components and structures. A reduction in bending capacity as well as changes in failure mechanism from flexural to bond are reported [10,11]. Rupture of the rebar is possible at high corrosion levels due to the formation of deep pits.

Corrosion in RC is typically a slow process. Therefore, it can take years before the first signs of concrete degradation can visually be observed, leading to large financial costs for maintenance and repair. An accurate assessment of the actual damage accumulation would allow a more accurate estimation of the residual capacity. In addition, early detection of damage progress with NDT can assist in preventive maintenance schemes which can possibly enable prolonging the service life of the structure. Presently, existing RC structures are typically assessed by visual inspection, core drilling, and electrochemical techniques [12]. These inspection techniques are valuable; however, they have several disadvantages such as being inspector-dependent, local, or destructive. Furthermore, they do not allow the continuous monitoring of the structure. Therefore, only sporadic inspections can be performed. The acoustic emission (AE) technique may serve as an important addition as it has four important advantages. First, it enables the continuous detection of damage in a non-invasive way by placing sensors on the surface of the structure. Second, as the signal is caused by the fracture process itself, an early detection of ongoing damage processes is possible. Third, it allows the characterization of the damage process as the characteristics of the AE signal are source-dependent. Fourth, the damage source can be localized based on the arrival time observed at different sensors.

AE-based detection and characterization of corrosion damage in RC have been widely studied. AE monitoring has been performed during the corrosion process as well as during mechanical testing of corroded RC components. Both parameter-based analysis as signal-based analysis have been discussed. Several researchers reported that the corrosion process can be divided in different stages based on the cumulative amount of AE events [13–16] and cumulative amount of AE energy [13,17]. These stages can be related to the onset of corrosion and nucleation of corrosion cracks. The parameters rise angle (RA), and average frequency (AF) are applied to determine the crack type (tensile and shear cracks). Ohtsu and Tomoda [15] reported high RA values and low AF at the beginning of the test representing shear cracking. Later, RA values decreased, and AF increased meaning that tensile cracking has initiated. However, crack width measurements were not reported. Kawasaki et al. [14] on the other hand, reported an increase of the RA value and a decrease of the AF when cracking was observed. Alternative to the parameter-based RA-AF analysis described before, a difference in crack type can also be obtained by a moment tensor analysis [14,18,19]. Only a few studies have been performed regarding the characterization of AE sources during the corrosion process [20,21]. A recent study by Van Steen et al. [22] reported a decrease of the peak and center frequency at the onset of concrete macro-cracking. AE sources, such as corrosion and concrete cracking, could be distinguished after analysis of the wavelet transforms of the AE signals.

Source localization of corrosion damage in RC remains challenging and is therefore less discussed in the literature. Proper localization is difficult as the P-wave velocity in steel is around 1.5 times the velocity in the concrete matrix. The wave can also be attenuated and reflected differently by both materials. Additionally, cracks have an important influence on the result [23]. The accuracy of the localization result depends on several parameters: the

sensor layout, the accuracy of the arrival time picking, the velocity model, and the propagation path of the wave. Several localization algorithms are available in the literature [24,25]. Most algorithms assume a homogeneous velocity and a straight propagation path. More complex algorithms allowing more realistic velocity models and propagation paths are still computationally very demanding. For a recent study on localization algorithms based on heterogeneous velocity models and fastest propagation paths, the reader is referred to Gollob et al. [26]. A drawback of these algorithms is that the gain in accuracy does not always outweigh the computational cost.

The aim of this paper is to localize internal damage in RC during the corrosion process with the AE technique. The localization result was improved by the development of a post-processing protocol, allowing the performance of dedicated filtering of the AE events. The protocol is applied and validated on an extensive set of experimental data. To investigate the applicability of the developed post-processing protocol towards on-site application, the methodology is scaled up using three sample scales: small mortar samples (scale 1), RC prisms (scale 2), and RC beams (scale 3).

## 2. Improvement of Post-Processing and Localization

To improve the AE source location results in RC, a post-processing protocol was developed and implemented in Matlab. A flowchart of the protocol is shown in Figure 1. Each step will be explained in the following subsections.

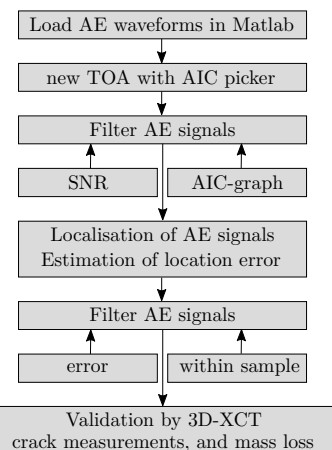

**Figure 1.** Flowchart of the developed AE post-processing protocol.

### 2.1. Arrival Time Picking

By an accurate determination of the arrival time, localization errors can be reduced significantly. Traditionally, the time of arrival (TOA) is defined by the first amplitude threshold crossing. However, this approach is sensitive to errors when this threshold is not set to an optimal value. In the presented experimental program, a first estimation of the TOA, $t_{\text{first}}$, was used as a trigger to discretize different signals and store them, together with a pre-trigger part of the signal.

A more exact estimation of the arrival time was subsequently determined using the Akaike Information Criterion or AIC picker as applied in the literature [27,28]:

$$\text{AIC}(t_{\text{w}}) = t_{\text{w}} \cdot \log_{10}(\text{var}(R_{\text{w}}(1, t_{\text{w}}))) + (T_{\text{w}} - t_{\text{w}} - 1) \cdot \log_{10}(\text{var}(R_{\text{w}}(t_{\text{w}} + 1, T_{\text{w}}))), \quad (1)$$

The signal $R_{\text{w}}$ is divided into two sections at a point $t_{\text{w}}$. Point $t_{\text{w}}$ is varied from 1 to $T_{\text{w}}$, with $T_{\text{w}}$ the non-dimensional length of the time window. In this paper, the time at which the maximum amplitude of the signal was reached, was chosen as $T_{\text{w}}$. The term $\text{var}(R_{\text{w}}(1, t_{\text{w}}))$ is the variance function of all samples from 1 to $t_{\text{w}}$ and $\text{var}(R_{\text{w}}(t_{\text{w}} + 1, T_{\text{w}}))$

is the variance function of all samples from point $t_w + 1$ to $T_w$. The absolute minimum of all values indicates the new onset time, $t_{AIC}$:

$$t_{AIC} = \underset{t_w}{\mathrm{argmin}}\, \mathrm{AIC}(t_w) \tag{2}$$

Figure 2 exemplifies the difference between the AIC picker and a threshold-based arrival time picker.

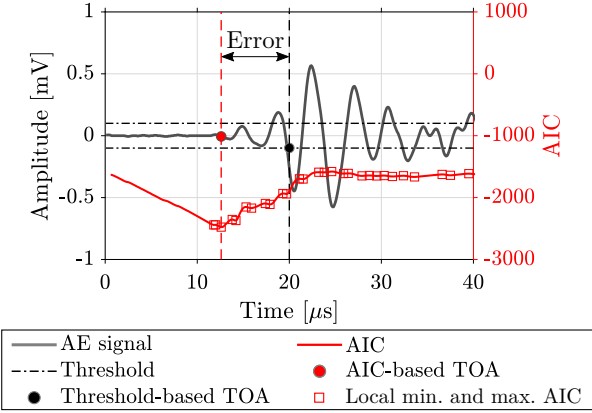

**Figure 2.** Difference between threshold-based TOA picking and AIC-based TOA picking.

AIC is more reliable than a threshold-based TOA picking; however, errors may still occur [29]. Two criteria were applied to filter the most reliable AE signals for which the TOA could be picked accurately. The first criterion is based on the signal-to-noise ratio (SNR) and evaluates whether there is the potential that the TOA can be accurately picked. The second criterion evaluates whether the TOA was accurately picked by AIC based on the shape of the AIC value graph [30].

The SNR is defined as the ratio of the signal power $S$ to the noise power $N$ in which $S$ is the maximum absolute amplitude value in mV of the entire signal, and $N$ is the maximum absolute amplitude value in mV of the first couple of microseconds of the signal (Equation (3)). The SNR threshold was set to 10, which was determined based on careful analysis of the recorded AE signals. If the SNR was larger than 10, the difference between the actual signal and noise level was large enough and the signals were kept for further analysis. A similar approach based on the root mean square was previously presented by Benavent-Climent et al. [31].

$$SNR = 20 \cdot \log_{10}\left(\frac{S}{N}\right) \tag{3}$$

The second criterion comprises the shape of the AIC value graph and has been presented by Gollob [30]. The criterion was adapted to the AE settings used in the current research. The actual TOA was estimated accurately if $t_{AIC}$ coincided with the first low point (local minimum) of the AIC graph. If this was not the case, the time difference between this first low point and $t_{AIC}$ was checked. It was found that if the time difference was less than 10 µs, the first local minimum was a better estimation of the arrival time than the global minimum. If the time difference was larger than 10 µs but the slope between the first low point and $t_{AIC}$ was smaller than $-25\,\frac{1}{\mu s}$, the estimation with the AIC picker was still found to be reasonably accurate. These limits were chosen based on a comparison with manually picked arrival times.

Both criteria needed to be fulfilled, otherwise the signal was excluded from the analysis.

### 2.2. AE Source Localization

Geiger's method was applied for AE source localization in which the best approximation of the source location is computed iteratively based on a least-squares approach. For a detailed description of the algorithm, the reader is referred to Ge [25]. In this algorithm, the wave velocity is considered to be homogeneous, and a straight propagation path is assumed between the source and the sensor.

A trial hypocenter $(x_0, y_0, z_0, t_0)$ is needed as a first guess of the source location. In this research, the middle of the rebar was chosen as a first hypocenter. The arrival times for each sensor can be calculated as:

$$t_{a,i} = t_0 + \frac{1}{v_p} \sqrt{(x_i - x_0)^2 + (y_i - y_0)^2 + (z_i - z_0)^2} \tag{4}$$

with $x_0$, $y_0$ and $z_0$ being the coordinates of an AE source, i.e., trial hypocenter, $t_0$ the onset time of the source, $x_i$, $y_i$ and $z_i$ the coordinates of the $i$th sensor, and $v_p$ the wave velocity. The wave velocity is determined during calibration before the start of a test.

As the trial hypocenter is a first guess, the calculated arrival times are different from the observed ones. The arrival time residuals ($\gamma_i$) can then be calculated for each sensor and should be minimized by applying correction factors ($\delta x$, $\delta y$, $\delta z$, and $\delta t$). If the corrections are relatively small, the arrival time function can be linearized and can be approximated as a first-order Taylor expansion:

$$\begin{bmatrix} \gamma_1 \\ \gamma_2 \\ \vdots \\ \gamma_m \end{bmatrix} = \begin{bmatrix} \frac{\partial f_1}{\partial x} & \frac{\partial f_1}{\partial y} & \frac{\partial f_1}{\partial z} & \frac{\partial f_1}{\partial t} \\ \frac{\partial f_2}{\partial x} & \frac{\partial f_2}{\partial y} & \frac{\partial f_2}{\partial z} & \frac{\partial f_2}{\partial t} \\ \vdots & \vdots & \vdots & \vdots \\ \frac{\partial f_m}{\partial x} & \frac{\partial f_m}{\partial y} & \frac{\partial f_m}{\partial z} & \frac{\partial f_m}{\partial t} \end{bmatrix} \cdot \begin{bmatrix} \delta x \\ \delta y \\ \delta z \\ \delta t \end{bmatrix}$$

or short:

$$\gamma = \mathbf{AB} \tag{5}$$

The Moore-Penrose inverse is applied to determine the correction factors:

$$\mathbf{B} = (\mathbf{A^T A})^{-1} \mathbf{A^T} \gamma \tag{6}$$

At the beginning of each iteration step, the trial solution is updated by adding the correction factor from the previous iteration step.

The process is repeated until two stopping criteria are reached. In this research, the first criterion is based on the size of the correction vector. When this correction vector goes to zero, it is a sign of convergency. In this research, a tolerance of 0.1 mm was chosen. The second stopping criterion is that the maximum number of iterations is set to 100.

Error ellipsoids can be used to visualize an estimation of the localization accuracy. Therefore, the covariance matrix can be calculated [29]:

$$\mathbf{C} = \sigma_d^2 (\mathbf{A^T A})^{-1} \tag{7}$$

where $\sigma_d^2$ is the data variance. As only spatial errors are of interest, $\mathbf{C}$ is a $3 \times 3$ matrix with the variances of the source parameters in direction of the three coordinates on its diagonal. The length of the semi-axis and the orientation of the ellipsoid can be determined from the eigenvalues $\lambda_i$ and the eigenvectors $v_i$ respectively. The length of the semiaxes, $l_i$, for a 68%-ellipsoid is determined as:

$$l_i = \sqrt{3.53\lambda_i} \tag{8}$$

The data variance $\sigma_d$ is usually unknown. In this research, following expression is used as presented by Schechinger [32]:

$$\sigma_{\hat{d}}^2 = \frac{\sum\limits_{i=1}^{m} \gamma_i^{\mathrm{T}} \gamma_i}{m} \tag{9}$$

with *m* the amount of observed arrival times.

Signals can be filtered based on a desired accuracy, meaning that a threshold could be set on the error (i.e., radius of the error ellipsoid). The protocol will also neglect signals of which the source is located outside the boundaries of the sample.

## 3. Experimental Test Program and Results

An extensive experimental test program was executed to gather AE data during the corrosion process of RC components, ranging from small mortar samples to RC beams. AE monitoring was performed with 2D and 3D sensor setups to validate the AE post-processing and source localization protocol. This section presents the results of the first two scales: a small mortar sample and two RC prisms. In the next section, the setup and protocol are upscaled towards RC beams.

### 3.1. Corrosion Damage Localization in a Small Mortar Sample

A cylindrical reinforced mortar sample with a diameter of 36 mm and a height of 60 mm was continuously monitored with the AE technique during an accelerated corrosion process. A smooth rebar with a diameter of 6 mm of which 2 cm was exposed to corrosion was positioned in the middle of the sample. The corrosion process was accelerated by imposing a direct current with a current density of 100 $\mu A/cm^2$. The rebar was connected to the positive side acting as the anode, whereas a stainless steel plate was connected to the negative side acting as the cathode. The sample was partially immersed in a 5% sodium chloride solution.

Six piezoelectric broadband sensors (50–2000 kHz) were attached to the sample with hot melt glue. The AE signals were amplified with preamplifiers with a gain of 34 dB. A Vallen AMSY-6 system was used for acquisition. The detection of background noise was prevented by setting an amplitude threshold of 38.5 dB. The pre-trigger time, duration discrimination time, and rearm time were set to 20 µs, 400 µs, and 3 ms, respectively. The sampling rate was set to 10 MHz. With 2048 samples being stored, the length of the stored signal was 204.8 µs. The digital frequency filter was set between 230 and 850 kHz. X-ray images were made at intermediate stages during the accelerated corrosion process, aiming at 3D X-ray microfocus Computed Tomography (3D-XCT) to visualize the internal damage sources.

A schematic representation of the setup is shown in Figure 3. More details of the setup can be found in Van Steen et al. [33].

Unfiltered localization results are shown in Figure 4. The localized AE events are plotted on X-ray images taken during the corrosion process. It allows a visual confirmation of the AE source localization results. The initial result is fairly well localized when compared to the observed crack and corrosion products. As monitoring was performed at higher frequencies, the error between the threshold crossing and actual TOA will be small, therefore resulting in a fairly good localization result even without additional post-processing and filtering.

Figure 5 shows the result after application of the post-processing protocol. The events that were localized by hits recorded by 4 sensors are indicated with a different marker as the events localized by hits recorded by 5 or 6 sensors. For the former, a unique solution exists, and no information is thus available on the localization error. For the latter, the error ellipsoids could be calculated. Here, the radius of the error ellipsoid was chosen to be maximum 2.5 mm in every direction. After post-processing, the localization result has been improved resulting in more events being localized near the crack and corroding zone,

which could again be verified with the X-ray scans. Most events are localized at the bottom side of the rebar which was closest to the sodium chloride solution.

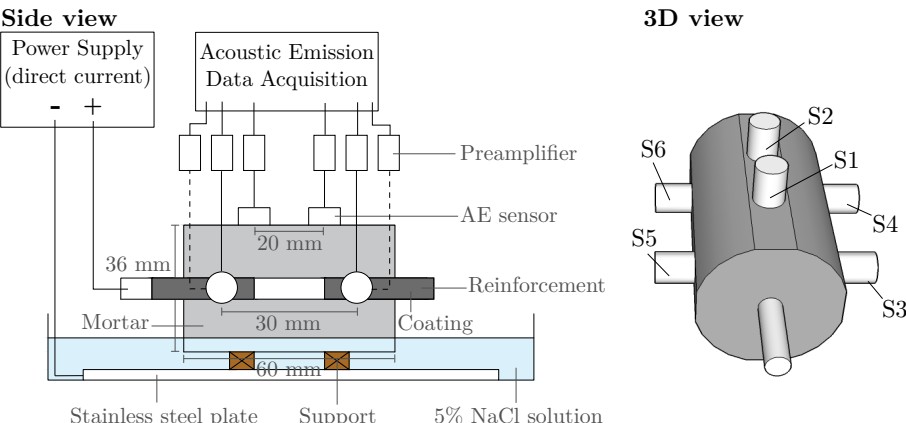

**Figure 3.** Schematic representation of the accelerated corrosion setup and 3D sensor configuration with six AE sensors (S1–S6) of scale 1.

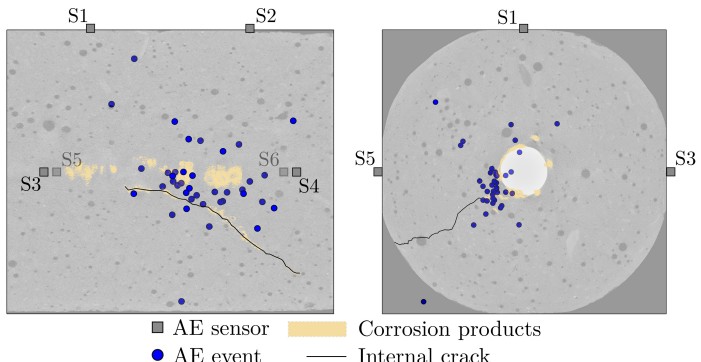

**Figure 4.** Unfiltered localized AE events; (**left**) view parallel to rebar, (**right**) view perpendicular to rebar.

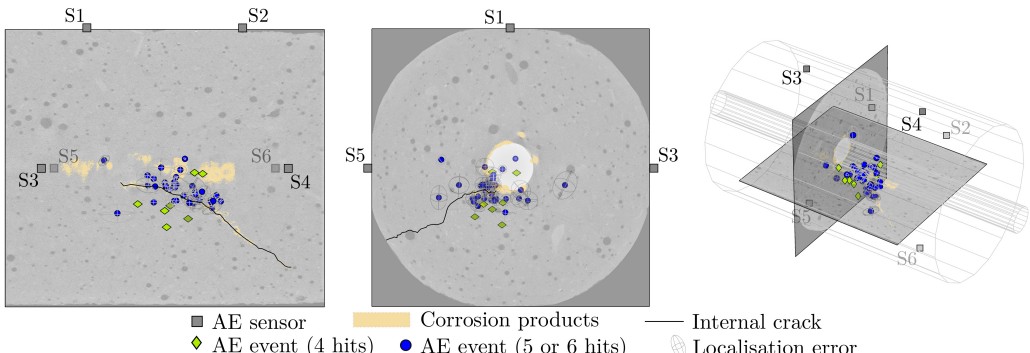

**Figure 5.** Localization results after applying the post-processing protocol with indication of the error ellipsoids, cracks, and corrosion products; (**left**) view parallel to rebar, (**middle**) view perpendicular to rebar, and (**right**) 3D view.

### 3.2. Corrosion Damage Localization in RC Prisms

The second sample scale contains two RC prisms with dimensions $150 \times 150 \times 250$ mm$^3$ that were monitored with a 3D AE sensor setup. The first sample was reinforced with one ribbed rebar with a nominal diameter of 14 mm in the center and will be referred to as AE-1. A length of 42 mm was bonded to the concrete and was able to corrode. The second sample was identical, but also had a stirrup with a diameter of 6 mm. The stirrup was placed around the main rebar and was connected to the mold with cable ties, following the setup

presented by Fang et al. [8]. The stirrup could corrode as it was connected with a copper wire to the main rebar. The second sample will be referred to as AE-2. The accelerated corrosion setup was scaled up from the mortar samples and is shown in Figure 6. A visual inspection was carried out every week during which crack width measurements were performed with a crack meter. The accuracy of the crack meter was 0.05 mm. The sample size was too large to allow for 3D-XCT analysis.

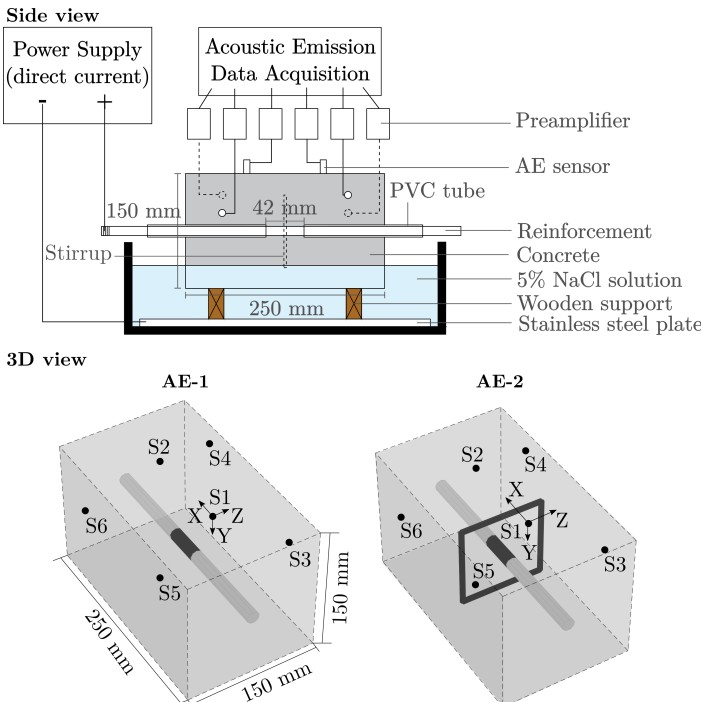

**Figure 6.** Schematic representation of the accelerated corrosion setup and 3D AE sensor configuration with six AE sensors (S1–S6) of scale 2.

The same AMSY-6 Vallen acquisition system with six channels was used. For this sample scale, broadband piezoelectric sensors with a flat frequency response between 100 and 400 kHz were attached on the specimen surface with hot melt glue. The sensor arrangement is shown in Figure 6. The amplitude threshold, the pre-trigger time, duration discrimination time, and rearm time were set to 40 dB, 20 µs, 400 µs, and 0.4 ms, respectively. The sampling rate was set to 10 MHz. With 2048 samples being stored, the length of the stored signal was 204.8 µs. The digital frequency filter was set between 95 and 850 kHz.

Figure 7 shows the AE source localization results without applying the post-processing protocol. The corroding part of the rebar is colored in dark grey. The bottom side of the rebar where most mass loss was observed after cleaning the rebar, is indicated in red. A mass loss of 4.03% was reached. Many reflections can be observed in the top part of the sample. The cracks on the bottom side and at the side of sensors 5 and 6 are not correctly localized. Please note that all AE sensors were positioned at the top half of the sample, as the bottom part was immersed in salt solution.

After applying the post-processing protocol, the AE localization result improved significantly as shown in Figure 8. The events that were localized by 4 sensors are again indicated with a different marker than the events localized by 5 or 6 sensors. For the latter, the error ellipsoids are also shown. The radius of the error ellipsoid was chosen to be maximum 10 mm in every direction.

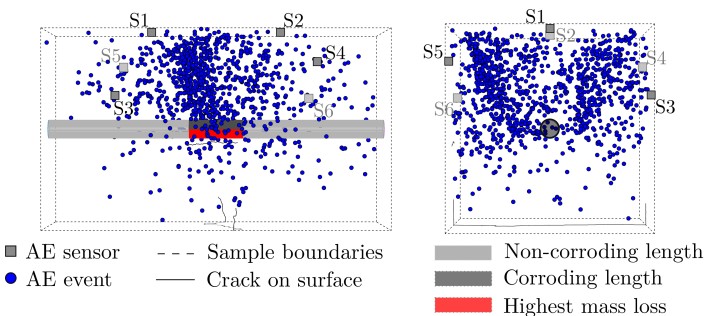

**Figure 7.** Unfiltered localized AE events of sample AE-1; (**left**) side view, and (**right**) front view.

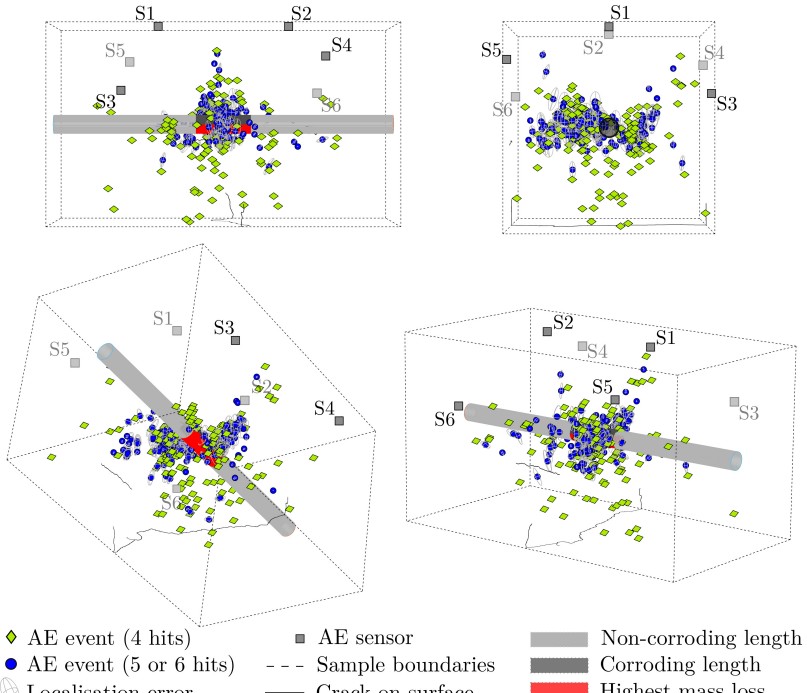

**Figure 8.** Localization results of sample AE-1 with indication of the error ellipsoids and cracks that were visible on the surface; (**top left**) side view, (**top right**) front view, (**bottom**) 3D views.

A crack, perpendicular to the rebar, could be noticed at the bottom side of the sample at the end of the experimental program. This crack is further denoted as crack 1. A second smaller crack at the side of sensors 5 and 6 and parallel to the rebar was observed. This crack will be further denoted as crack 2. It can be seen that most AE events are localized around the rebar, reaching the side of the sample where crack 2 was observed. Less AE events are localized near crack 1 due to the AE sensors being positioned on the top half of the sample.

Before the accelerated corrosion test, pencil lead breaks (PLB) were performed on the samples. PLBs were performed on all sides of the sample (including the bottom side), once when the sample was dry and once when the sample was partially saturated by the sodium chloride solution. It was found that the AE post-processing protocol was able to localize the PLBs executed at the bottom side of the sample. However, the localization error was found to be larger than AE signals originating from PLBs executed within the sensor array. Hence, less reliable source locations at the bottom of the sample will be filtered out during post-processing.

A time lapse of the localization result is shown in Figure 9. In the beginning of the test, most events are localized close to the rebar meaning that these events are caused by the corrosion process itself and micro-cracking caused by the expansion of corrosion products.

Afterwards, cracks start to grow from the inside of the sample towards the outside. It can be noticed that many AE events were localized before the cracks could be observed on the surface meaning that the AE technique is able to assess the internal damage earlier than a visual inspection.

The time lapse shows why crack 1 was less accurately localized. The cracks start to grow internally, near the rebar. Therefore, as this crack grows, this internal crack eventually prevents the AE signals originating from the bottom part to reach at least 4 sensors. Another sensor layout with sensors at the bottom would overcome this problem. However, also during on-site monitoring, sensors cannot always be positioned optimally, and hence dedicated post-processing is a necessity to avoid false detections and large localization errors.

It can be concluded that the initial internal crack was localized correctly. This is validated by comparison with the zone of the rebar that was most heavily damaged, i.e., most mass loss. As damage progresses, crack growth is more difficult to localize correctly due to increased heterogeneity of the concrete.

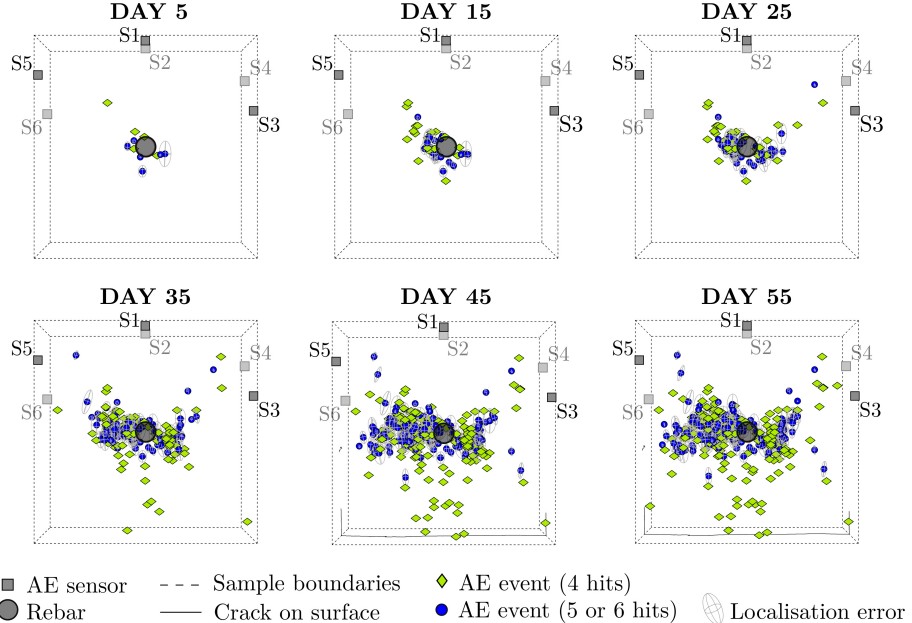

**Figure 9.** Time lapse of the localization results of sample AE-1.

The aim of sample AE-2 was to investigate whether corrosion of a stirrup could be localized correctly.

Figure 10 shows the AE source localization results without applying the post-processing protocol. The corroding parts of the rebar and stirrup are again colored in dark grey. The side of the stirrup where most mass loss was observed after cleaning, is indicated in red. The localization result is scattered and fails to provide accurate information on the most damaged part of the sample.

A small shrinkage crack could be observed before the start of the accelerated corrosion process. Figure 11 shows the localization result after post-processing. Most AE events are localized around the stirrup at the side where the crack was visible on the surface. The initial shrinkage crack was found to locally promote the chloride ingress, as also observed in the literature [34]. Therefore, the corrosion process likely initiated at this point. This was validated by removing and cleaning of the stirrup as shown in Figure 12. A pit can clearly be seen at the spot where the crack and AE events are located. A few AE events were localized around the main rebar. The mass loss of the main rebar was 1.54%, whereas the mass loss of the stirrup was 2.95%. The stirrup was clearly more heavily corroded than the main rebar which was correctly represented by the AE measurements after application of the developed post-processing tool.

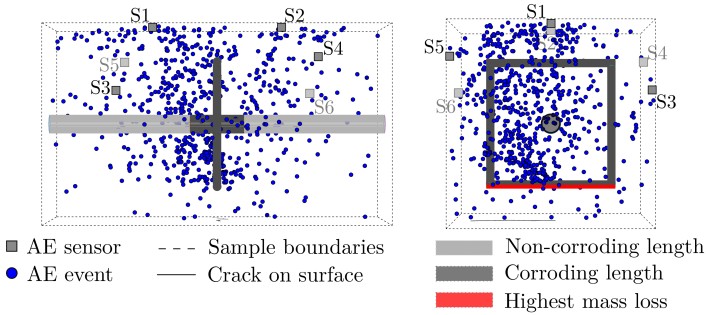

**Figure 10.** Unfiltered localized AE events of sample AE-2; (**left**) side view, and (**right**) front view.

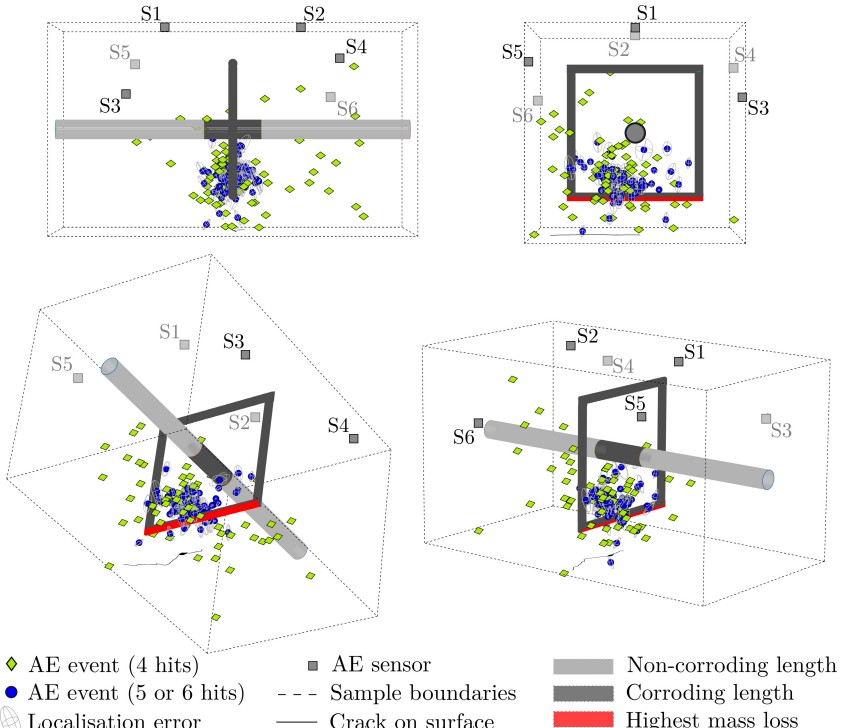

**Figure 11.** Localization results of sample AE-2 with indication of the error ellipsoids and cracks that were visible on the surface; (**top left**) side view, (**top right**) front view, (**bottom**) 3D views.

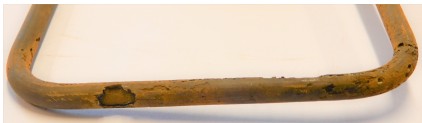

**Figure 12.** Picture of the stirrup of sample AE-2 after removal and cleaning at the end of the accelerated corrosion test.

## 4. Upscaling to RC Beams

From the previous section, it can be concluded that the AE technique is able to detect and localize damage due to reinforcement corrosion in 3D. As a next step, the protocol was scaled up towards RC beams. However, for this sample scale, a larger monitoring area was required. Therefore, a 2D setup was proposed.

Two RC beams with dimensions $150 \times 200 \times 1800$ mm$^3$ were monitored with the 2D setup. They were reinforced with a ribbed rebar with a nominal diameter of 14 mm. A length of 1500 mm was exposed to corrosion. The mold was designed in such a way that a concrete cover of 30 mm was obtained.

The corrosion process was accelerated using the same principle as for the mortar sample and RC prisms. Instead of immersing the beams partially in the sodium chloride solution, it was more convenient to place the beams upside-down. A bottomless wooden tank with a length of 1250 mm was placed on top of the samples and sealed with silicone. In this tank, the 5% sodium chloride solution was added. The current density was lowered to 50 μA/cm² to slow down cracking of the concrete cover. A visual inspection was carried out weekly during which crack width measurements were performed with a crack meter. The accuracy of the crack meter was 0.05 mm. A schematic representation of the setup is shown in Figure 13. More details on the corrosion setup can be found in Nasser et al. [10] and Van Steen et al. [22].

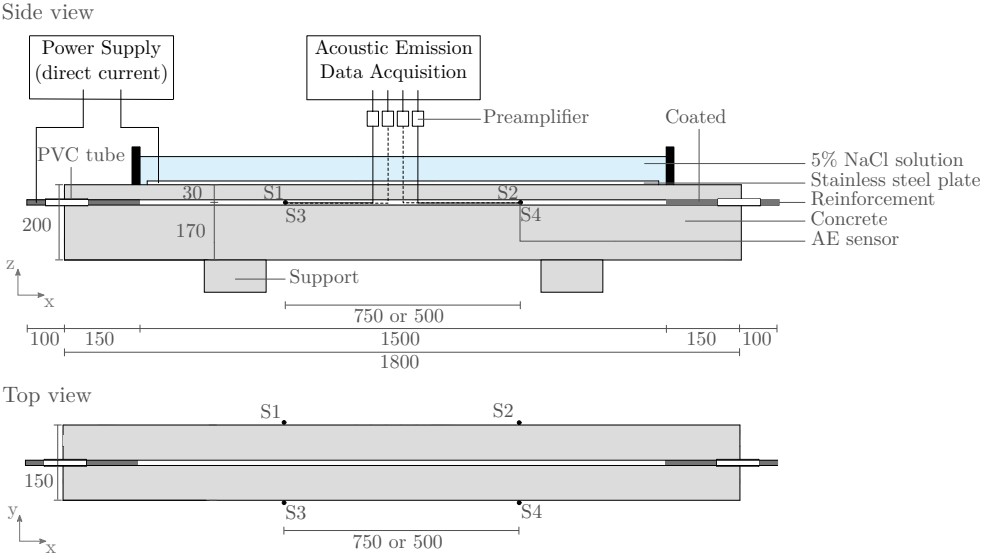

**Figure 13.** Schematic representation of the accelerated corrosion setup and 2D AE sensor configuration (S1–S4) of the RC beams.

The same AMSY-6 Vallen acquisition system with six channels was used. Broadband piezoelectric sensors with a flat frequency response between 100 and 400 kHz were attached with hot melt glue. A 2D sensor layout was used to monitor the beams continuously. Four sensors were attached to the concrete, two at each side of the beam (Figure 13). The distance between the sensors along the length of the beam was 750 and 500 mm for Beam-1 and Beam-2 respectively. The amplitude threshold, pre-trigger time, duration discrimination time, and rearm time were set to 40 dB, 200 μs, 450 μs, and 5 ms, respectively. As a longer pre-trigger time was used for the RC beams, namely 200 μs instead of 20 μs for scales 1 and 2, also the shape of the AIC curve changed. Therefore, the threshold for the slope to check whether the TOA was accurately picked was adapted from $-25 \frac{1}{\mu s}$ to $-5 \frac{1}{\mu s}$. The sampling rate was set to 5 MHz. With 4096 samples being stored, the total length of the signal was 819.2 μs. The digital frequency filter was set between 95 and 850 kHz.

Figure 14 shows the initial localization results of Beam-1. The result is shown on pictures of the top surface of the beam and the crack that was observed on the surface is indicated. Please note that the crack is exaggerated to make its position clearer. The localization result is scattered, and many events are localized outside the sensor array.

Figure 15 shows the filtered results obtained with the post-processing protocol. AE events are mainly localized around the crack; however, some events are also localized at the sides where no cracking on the surface was observed. Many events were recorded by three sensors, therefore, the localization error could not be calculated and used as a filter. This may be overcome by placing the sensors closer to each other. For events recorded by four sensors, the error was set to be maximum 25 mm (radius of the error ellipse).

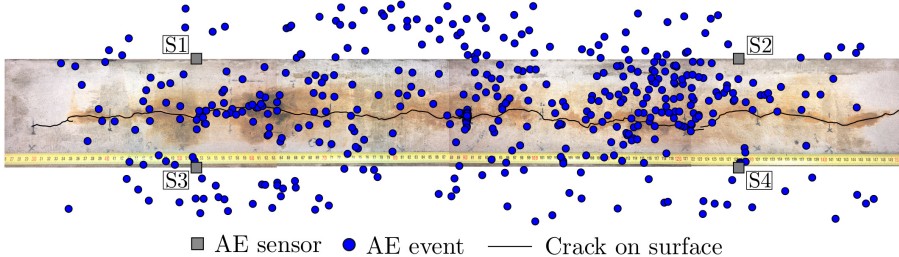

**Figure 14.** Top view of the unfiltered localized AE events of Beam-1 with indication of the crack that was visible on the surface.

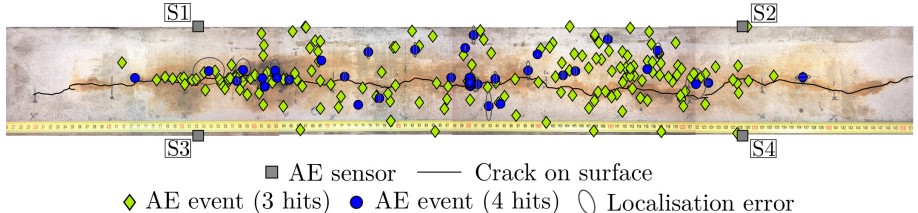

**Figure 15.** Top view of the localization results of Beam-1 with indication of the error ellipses and crack that was visible on the surface.

The localization result at different moments during the test is shown in Figure 16. The localized AE events are plotted on a picture taken when the crack measurements were performed. The measured crack widths are shown on top of each picture. Please note that the crack is exaggerated in the picture to make its position clearer.

On day 5, initial events were localized mainly at the left side of the beam. However, no crack was observed by visual inspection of the beam. On day 11, a crack could be seen at the left side of the beam as predicted by the AE results. Eight events were observed close to sensors 2 and 4; however, the crack did not reach this part of the beam yet. On day 21, the crack grew towards sensors 2 and 4. As the test continued, more events could be observed. From this time lapse, it is observed that the AE technique can predict where the damage will occur before the damage can be observed by a visual inspection. The technique can therefore give an earlier warning than a visual inspection, which was also observed on scale 2.

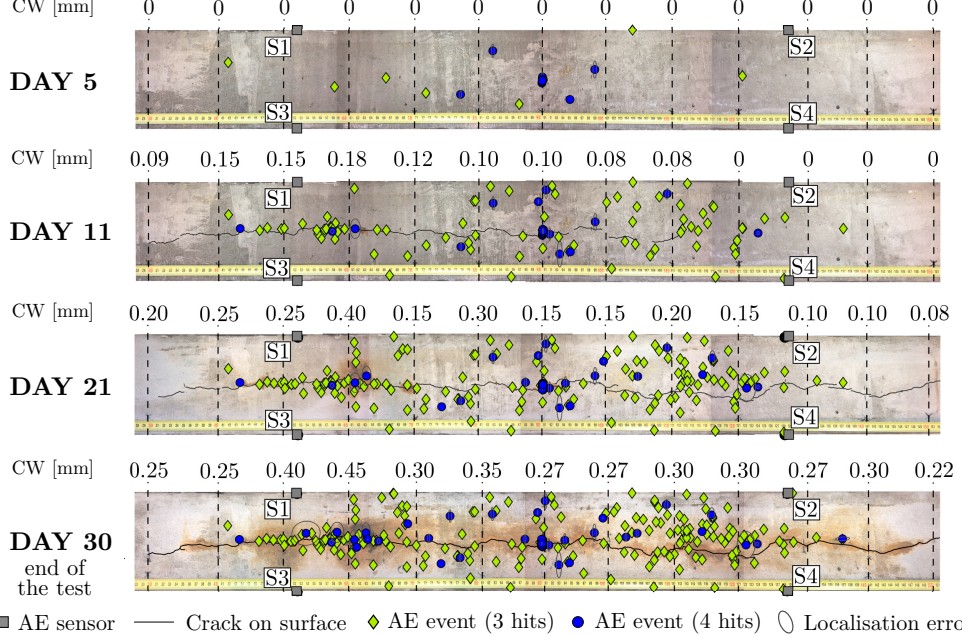

**Figure 16.** Time lapse of the localization results of Beam-1.

The unfiltered and filtered results of Beam-2 are respectively shown in Figures 17 and 18. The unfiltered results show many reflections at the position of the sensors. Moreover, many events are localized outside the sensor array. The improved result is shown in Figure 18. The same conclusions can be made as for Beam-1. Please note that more events could be localized by 4 sensors as the horizontal distance between the sensors is less, namely 500 mm instead of 750 mm.

In general, the localization error will be larger for the 2D setup of the beams than the 3D setup of scales 1 and 2. For scale 3, the sensors are placed on the same height as the rebar. Therefore, signals originating from corrosion and initial micro-cracking may be localized well. However, cracking of the concrete cover will start at the level of the rebar, but will grow towards the outside. As this is a 3D phenomenon monitored with a 2D setup, this will result in larger errors. Moreover, as the crack grows, a straight propagation path will no longer be valid as the concrete becomes more heterogeneous for wave propagation.

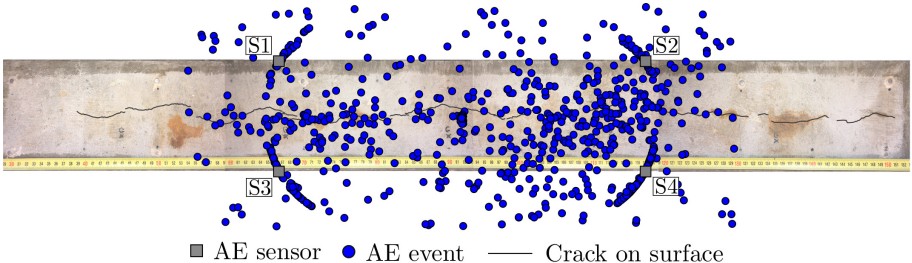

**Figure 17.** Top view of the unfiltered localized AE events of Beam-2 with indication the cracks that were visible on the surface.

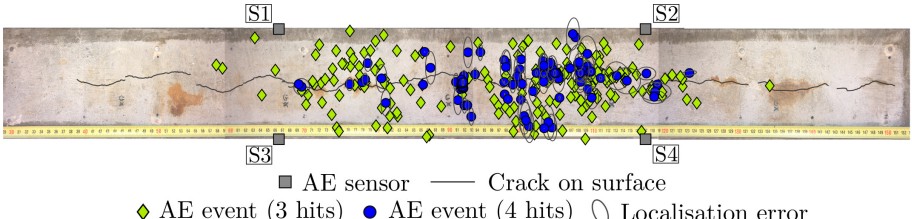

**Figure 18.** Top view of the localization results of Beam-2 with indication of the error ellipses and cracks that were visible on the surface.

## 5. Conclusions

This paper focused on AE source localization of damage due to corrosion in RC. Three sample scales were investigated: a small mortar sample (scale 1), RC prisms (scale 2), and RC beams (scale 3). For each scale, the corrosion process was accelerated by an imposed direct current.

It is found that the AE technique can detect corrosion in RC. However, it is observed that dedicated filtering is necessary to reliably localize AE events. For this purpose, a new post-processing protocol was developed by combining existing methods to improve the arrival time picking and filters. A better estimation of the wave's arrival time is obtained by applying the Akaike Information Criterion (AIC). Afterwards, signals are filtered based on their signal to-noise ratio (S/N) and accuracy of the arrival time picking. The novelty of the work lies in the validation of the developed protocol by means of extensive experimental results obtained during accelerated corrosion experiments. It is proven that the developed post-processing protocol significantly improves the localization result on all three sample scales.

On the smallest sample scales, scales 1 and 2, a 3D sensor layout was applied. It was found that 3D localization can localize initial internal cracking which could be verified with X-ray images (scale 1), crack measurements (scale 2), and determination of the mass loss (scale 2). As damage progresses, crack growth is more difficult to localize correctly. Localization in 3D may be valuable in a laboratory environment, e.g., to validate cluster

algorithms to differentiate AE sources. When larger areas need to be monitored, a 2D sensor layout may be more convenient. Therefore, for the RC beams (scale 3), a 2D sensor layout was used as the assessment of a larger monitoring area was desired. Localization results were verified by comparison with crack measurements. Corrosion and initial micro-cracking were localized correctly. A larger error is noticed during crack growth as this is a 3D phenomenon.

For all scales, it was found that post-processed AE results enabled the detection and localization of corrosion-induced damage correctly and earlier than a visual inspection. The developed protocol can serve as a valuable and reliable alternative to computationally demanding localization algorithms. The applied method allows the prediction of where the visual damage will be first observed. This advantage can be useful during on-site monitoring to provide well-informed decisions on maintenance and repair.

This paper presented both 2D and 3D localization setups which both have advantages and disadvantages. In actual structures, the extent of the measuring zone should be considered against the desired accuracy of the localization result. As a first assessment of the structure, sensors may be spread over a larger area and less sensors will be available for localization. A 2D sensor layout may be suitable to cover this larger area and have sufficient information to pinpoint the most damaged zones. Afterwards, the areas of interest can be monitored with a 3D sensor setup for a more detailed assessment. On-site application will be part of future research.

**Author Contributions:** Conceptualization, C.V.S. and E.V.; methodology, C.V.S. and E.V.; software, C.V.S.; validation, C.V.S.; formal analysis, C.V.S.; investigation, C.V.S.; resources, E.V.; data curation, C.V.S.; writing—original draft preparation, C.V.S.; writing—review and editing, C.V.S. and E.V.; visualization, C.V.S.; supervision, E.V.; project administration, E.V.; funding acquisition, C.V.S. and E.V. All authors have read and agreed to the published version of the manuscript.

**Funding:** This research was performed within the framework of project C24/17/042 "Multi-scale assessment of residual structural capacity of deteriorating reinforced concrete structures", supported by Internal Funds KU Leuven. Moreover, the financial support by FWO-Flanders for the postdoctoral mandate of C. Van Steen (Grant No. 12ZD221N) is gratefully acknowledged.

**Institutional Review Board Statement:** Not applicable.

**Informed Consent Statement:** Not applicable.

**Data Availability Statement:** Not applicable.

**Conflicts of Interest:** The authors declare no conflict of interest.

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
