# Peer review of "Degradation Monitoring in Reinforced Concrete with 3D Localization of Rebar Corrosion and Related Concrete Cracking"

_applsci, doi:10.3390/app11156772_

Round 1

Reviewer 1 Report

The manuscript "Degradation monitoring in reinforced concrete with 3D localisation of rebar corrosion and related concrete cracking" reports an interesting and original study on the use of the acoustic emission technique to monitoring the reinforced concrete structural elements subjected to corrosion phenomena. In particular, three different sample scales are considered in order to perform the laboratory tests: small mortar samples (scale 1), RC prisms (scale 2), and RC beams (scale 3). 

General consideration:

  • the topic of the paper is interesting especially for the attention it is giving to the existing concrete structures in recent years
  • the laboratory tests were well conducted and the results clearly exposed in the manuscript, even considering a critical point of view
  • the paper is well organized and well written

For these reasons it is opinion of this reviewer that the paper can be accepted for the publication in Applied Science Journal after the following minor improvements:

  • improve the introduction considering the corrosion effects on the bearing-capacity of the RC structures as roperted in: (1) Crespi P, Zucca M, Valente M. "On the collapse evaluation of existing RC bridges exposed to corrosion under horizontal loads". Engineering Failure Analysis, 116, 104727, 2020, (2)  Stochino, F., Fadda, M.L., Mistretta, F.
    Assessment of RC bridges integrity by means of low-cost investigations (2018) Frattura ed Integrita Strutturale, 12 (46), pp. 216-225
  • better clarify the originality of the results obtained in the conculsions
  • improve the References list

Reviewer 2 Report

According my opinion, this paper is well written. The format is very good. The paper is written clearly and good way.

The topic is interesting for community. The proofs are easy to follow and authors give good conclusions. They even extend the existing results and bring good new results. 

I recommend the publication of the paper under small corrections: Improve the background, including more references that are relevant.

Reviewer 3 Report

The manuscript is well writthen. The authors describe accuratly the problem and the solution they propose. The post processing method looks promising and the results depict that it work. The method could be useful even as it is in infrastracture that already exist. I would say that the subject is well covered although some more groups of researchers could be included in the introduction because they deal with similar problems of RC structures with the use of AE.
